# Growing Up with HIV: Experiences of Transition from Adolescence to Adulthood at Selected Primary Health Facilities in Limpopo Province, South Africa

**DOI:** 10.3390/children10050798

**Published:** 2023-04-28

**Authors:** Azwinndini Cecilia Mukwevho, Maria Sonto Maputle, Dorah Ursula Ramathuba

**Affiliations:** Department of Advanced Nursing, University of Venda, Private Bag X5050, Thohoyandou 0950, South Africa; cecilianndini@gmail.com (A.C.M.); dorah.ramathuba@univen.ac.za (D.U.R.)

**Keywords:** adolescents living with HIV, disclosure, transition to adulthood, non-adherence

## Abstract

Background: Many children who contracted Human Immunodeficiency Virus (HIV) through vertical transmission are now in their adolescent and early adult years. The aim was to explore the experiences of adolescents living with HIV (ALWHIV) during the transition from childhood to adulthood. Methods and Material: The study was conducted at selected primary healthcare facilities in the Mopani and Vhembe districts in July 2021. A qualitative research approach that included contextual, descriptive, and exploratory designs was used. The population comprised 27 ALWHIV who were purposively sampled and enrolled for ART care. Data were collected using in-depth interviews, and the question was *“How is it for you as you live with a virus and transit from adolescent to adulthood”.* The open coding approach was used to analyse the data. Measures to ensure trustworthiness articulated in Lincoln and Guba’s criteria and ethical considerations were adhered to. Findings: The findings revealed four themes: poor understanding of the disease condition, improved physical health when adhering to ARV treatments, challenges related to sexual maturity and intimate relationships, and parents not disclosing their children’s HIV status. Conclusion: Parents’ delayed and non-disclosure of adolescents’ positive HIV status led to a lack of awareness about the course of the disease, non-adherence to ART, and unsafe sex practices that could increase the risk of HIV transmission and re-infection. To address these multiple obstacles associated with ALWHIV, a comprehensive, multi-sectoral approach that is teenager-friendly should be undertaken.

## 1. Introduction

The number of adolescents with perinatally acquired Human Immunodeficiency Virus (HIV) has increased in low-income countries, especially in sub-Saharan Africa, where HIV prevalence and incidence are the highest [1]. In 21 of the 22 target countries for the Global Plan, the percentage of pregnant women with HIV receiving antiretroviral treatment (ART) increased from 36% in 2009 to 80% in 2015 [2]. The risk of mother-to-child transmission (MTCT) can be reduced to under 5% when using ART and other Prevention of Mother to Child Transmission (PMTCT) therapies [2]. Since 1995, using PMTCT strategies has helped to avert almost 35,600,000 new HIV infections among children. Despite this tremendous advancement, in 2015, 150,000 children (400 children per day) contracted HIV, and 23% of HIV-positive pregnant mothers lacked adequate access to ART. Most of these children have grown to be adolescents living with HIV (ALWHIV) who were vertically infected [3]. Estimates show that in the past five years, from 2010 to 2015, 1.3 million of these should have been avoided [2].

An HIV-positive woman’s baby is at risk of contracting the virus during pregnancy, childbirth, and breastfeeding [1,2,4]. The availability of antiretroviral drugs (ARVs) in public institutions and the fact that counselling and HIV testing services are free for everybody have impacted the day-to-day life of all people worldwide. Adolescents living with HIV and receiving ART reportedly have an increased likelihood of poorer treatment adherence and viral suppression compared to adults receiving ART, who achieve higher rates of viral suppression [5,6]. There is a pressing need to move adolescents from pediatric to adult care in the era of ART. This could be achieved as the scaled-up measures of free and easy accessibility to ART. However, HIV incidence in children from developing countries has shown an appreciable decline [7]. Teenagers are most likely to disappear or stop taking their medication during this transitional period, which has a negative effect on clinical outcomes [8]. In the case of teenage girls, this can raise the risk of pregnancy and transmission to partners [9]. The primary goal of HIV treatment is to reach undetectable viral loads. This facilitates immune system recovery, lowers the risk of AIDS-related diseases and mortality, and lowers the likelihood of HIV transmission to others [10]. The World Health Organization (WHO) recommended viral load monitoring as the gold-standard mechanism for keeping track of the effectiveness of HIV treatment in 2013 [11]. PMTCT programmes offer ART to HIV+ pregnant women to stop the spread of the virus to their unborn children. The difficulties adolescents have in taking their medication result from several factors, including behaviour, lack of disclosure, stigma, psychosocial support, and childhood forgetfulness. These factors further include age, developmental stage, type of caregiver, and support [7].

With the programme of Test and Treat, which was rolled out in 2017, all pregnant women who tested positive were enrolled in the ART programme as soon as possible. This helps women to know their HIV status earlier [12,13,14]. They were given thorough education concerning medication, adherence, compliance, and the importance of using condoms during sexual intercourse for the baby’s sake in utero. If there were no complications, the woman was given the medications; if there were other complications such as tuberculosis (TB), ART was delayed for two weeks following clinical investigations [15]. HIV transmission from mother to child without treatment has a 15% to 45% chance. Hence, some babies born from HIV-positive mothers contracted the virus. Most of these adolescents had vertical infections [4]. However, ART and other successful PMTCT measures lowered this risk to under 5% [16].

ALWHIV face many more challenges than those without the disease. Amid strict parental consent laws and regulations [17], ALWHIV still undergo physical, psychological, and sexual development; they regularly deal with complex developmental, psychological, and sex concerns (Lowenthal et al.) [18]. They should often be involved in decision making, policies, and programmes, which involve their needs to have a sense of belonging. Extra support is of great value to these adolescents to ensure they can engage in these processes meaningfully [19]. Many parents or guardians neglect to disclose their children’s HIV status to them as they grow up, which may reflect a lack of commitment to their care [20]. Therefore, these adolescents might have unanswered questions, which make them non-adherent to their medications. Growing up was linked to a deterioration or reversal of immunological healing [21]. Adolescents in transition often have an unreduced viral load, high attrition rates, and loss to follow-up [14]. ALWHIV live in fear as there is still no cure for the disease. They also fear being stigmatized by those who know their status or that of their parents [22]. These ALWHIV were also afraid to visit health facilities when sick as they were uncertain of the confidentiality and fear of stigmatization by health providers. Adolescents should be informed of their health status instead of being stigmatized by health-care providers. Therefore, self-disclosure needs to be intensified because children with vertically acquired HIV are becoming adolescents and are faced with sexual life challenges. The need to be informed includes family planning, sexually transmitted diseases, and condom negotiation [23,24]. Ruria et al. [25] concurred with the above notions that sexual and reproductive health needs, the effect of trauma on their HIV status, mental health needs, and the impact of chronic physical illness, among other issues, must be addressed [26]. Stangl and Sievwright [27,28] further revealed that adolescents have questions about biopsychological changes related to their development.

As a clinical nurse practitioner in one of the ARV clinics in Collins Chabane municipality of Vhembe district, the researcher noticed that adolescents diagnosed with HIV present with anger and have unanswered questions such as: *“Why did this happen to me?”* Thus, they often default on their appointments for treatment collection once they transition from childhood to adulthood. In 2017/2018, amongst 24.8% of youth aged 15 to 19 who tested positive, 5.5% were pregnant in Limpopo province [29]. Some sexually active adolescents have unprotected sex without telling their partners they have HIV. This could increase the rate of HIV transmission and re-infection and they are at risk of premature death. The study aimed to explore the experiences of ALWHIV during the transition from childhood to adulthood.

## 2. Methods and Material

A qualitative approach, wherein exploratory, descriptive, and contextual designs were employed, was undertaken. The research was conducted in July 2021 at nine health facilities in selected primary health-care (PHC) facilities in Mopani and Vhembe districts at Thulamela, Makhado, and Greater Giyani municipalities because of the high teenage pregnancy rate (as a segment of the doctoral study).

### 2.1. Population and Sampling

All ALWHIV visiting the ARV monitoring clinic made up the population. This included adolescents aged 12–19 years who tested HIV+ and were enrolled on the ART programme, and were on ART long-term medications and disclosed HIV status. The exclusion criteria were adolescents who are HIV-positive but who have not enrolled on an ARV programme. Twenty-seven (27) teenagers who volunteered to take part were purposefully sampled.

### 2.2. Ethical Considerations

Researchers first obtained ethical approval from the University of Venda Ethics Committee (Ref: SHS/20/PH/27/3009), authorization to conduct the study from the Limpopo Provincial Department of Health (LP 2020 10 028), and managers of District Primary Health-Care institutions (S5/6) before gathering data. After receiving the minor’s full disclosure, parents or legal guardians provided written consent, and ALWHIV also provided written assent. Participants were supplied with information regarding the goal and scope of the study and how the results would be used. Privacy, anonymity, confidentiality, and rights to self-determination were adhered to. They were advised that participation was entirely voluntary and could stop participating in the study without facing any repercussions.

### 2.3. Data Collection

In-depth, unstructured individual face-to-face interviews were used to gather data for a thorough narrative on their experiences living with the virus as they transitioned from childhood to adulthood. The researcher asked the unit manager to assign a quiet area so the interviewees could feel comfortable and private. Central opening questions sparked the discussion: “How is it for you as you live with a virus and transit from adolescence to adulthood*”.* Researchers conducted the interviews at the health facility using the participants’ local language (TshiVhenda or Xitsonga), each lasting 30 to 45 min. Participants gave their consent for the voice recorder to be used.

### 2.4. Data Analysis

Data analysis began while data were being collected, not once data collection was complete. Qualitative researchers start their analysis early on when collecting data for a project. Tesch’s 8-step open-coding data analysis process was employed [30]. A linguist converted the data from Tshivenda/Xitsonga to English verbatim. To avoid omissions, transcriptions of the data were compared to the recorded information. To establish a connection between the earlier studies and the current research, literature control was carried out to guarantee that the data and pertinent literature fit well together.

### 2.5. Trustworthiness

The Polit and Beck [31] trustworthiness criteria were adopted to ensure trustworthiness. The prolonged engagement ensured credibility. The researcher scheduled a meeting with the participants to build rapport. The researcher spoke with the interviewees for a considerable time while also listening to and observing them. Interviews with the participants continued until data saturation was reached. To verify the accuracy of the information and the conclusions, a member check was also conducted. To ensure authenticity, the voice recorder was utilized. Detailed descriptions of the research approach ensured transferability. The verbatim transcriptions of the recorded interviews contained nonverbal cues (such as silence/sighs, frowns, and leaning back) in brackets to confirm their veracity.

## 3. Presentation of Findings

### 3.1. Profile of ALWHIV (n = 27)

Table 1 present demographic profile of 27 ALWHIV who participated in the study.

### 3.2. Themes Reflecting the Experiences of ALWHIV

Four themes emerged, as presented in Table 2.

Participants’ direct quotes were italicized and classified according to district and numbering code.

#### 3.2.1. Theme 1: Poor Understanding of the Disease Progression

The ALWHIV mentioned a lot of challenges that they have experienced from childhood to adolescence. Further, participants displayed poor knowledge and understanding of the disease process, as cited.

Participant 05 from Shayandima confirmed, ‘*I didn’t know what was wrong with me, and I was taken to prophets, traditional healers, and doctors, but all in vain. Until I was admitted, and my parents consented that I be taken blood to check my HIV status. The results came back positive*’.

Participant 01 from Makonde revealed, ‘*my childhood has been harrowing and demanding from my mom and grandma because I was always sick. I had to visit the hospital now and then I was always admitted they were saying I am being bewitched. But I was stable after testing and starting ARVs*.

Participant 15 from Ntluri shared, ‘*I grew up being sick and was diagnosed very late because my family believed in traditional practices and medicines. But they were ineffective to the virus, I was HIV+ by then I was thirteen, and understanding how and what HIV means to me was a problem*’.

However, one participant displayed understanding when suspecting a grandchild had contracted the virus from the late mother.

Participant 03 from Shayandima said, ‘*my granny said when I was seven months old my mom died. Two years later, I became very sick. Granny asked the nurses that I be tested for HIV. The results came back positive, meaning I was born with the virus. That’s how I knew about my status. From there, I’m drinking ARVs till to date*’.

According to UNICEF [1], everyone should have a thorough grasp of HIV to have a positive relationship with their medical condition if they have the disease.

#### 3.2.2. Theme 2: Improvement in Physical Health Experienced as Associated with Adherence to Treatments

The study findings showed that ALWHIV, as they grow up, become mature and responsible. The direct quotes from participants to confirm this were:

Participant 04 from Shayandima said, ‘*When the phone beeps on the wall, I always know it’s time for my medicine and if I’m with my cousins or we are studying, they will look at me or bring me water to take my tablets and life continues am healthy now*.

Participant 06 from Madombidzha said, ‘*Unfortunately, my body was not taking medications well for a long time (virological failure). They checked my blood every two months, and I changed my medications. The doctor told my aunt he would switch me off these medications and try other antiretroviral drugs. Then I was given new medications. From there, I started to be well and gain weight*’.

Participant 20 from Makhuvha corroborated the importance of adherence when saying, ‘*my grandmother always said don’t forget your appointment day and the remaining pills. Pill counting was always done and when I returned, I would give her everything; she is very strict, so she makes sure I swallow the tablets every time*”.

Participant 27 from Tshakhuma also confirmed, ‘*I must always drink these tablets/medications because they are my life whether I like it or not, they have changed my life completely. The way I suffered during my childhood before I started medications was too much*’.

Dennison et al. [32] said that participants need to be empowered with knowledge, especially adherence to treatment, to make informed decisions.

#### 3.2.3. Theme 3: Challenges Related to Sexual Maturity and Intimate Relationships

As ALWHIV grow up, they may engage in a relationship with a partner of the opposite sex. They might be afraid to disclose their status to their partners. This was found to be the case as in this study. When ALWHIV discovered they had been misled about their HIV status, they said they felt angry and resentful; they mentioned the following.

Participant 06 from Makhuva explained, *‘my parents told me that having this disease is secretive. Don’t share with anyone at school or your partner because they will mock, harass, and isolate me*’.

Participant 04 from Muila said, ‘*I knew about my status when my girlfriend fell pregnant and told me she tested positive at the clinic when starting antenatal care (ANC). At first, I was angry at her and told her she had given me the disease from her previous relationship. She was my first girlfriend, and I was not the first to her and I couldn’t cope and started abusing alcohol*.’

Participant 16 from Ntluri said, ‘*I was told not to engage to unprotected sex, but no reason was given to me and that made me curious and to engage. After knowing my status, I was angry because I was supposed to be told*’.

Disclosure is not a one-off event; communication must continue and be adapted as the child grows into adolescence and early adulthood. This will pave the way to developing a good support network in many spheres of life [33].

#### 3.2.4. Theme 4: Lack of Disclosure of HIV+ Status by Parents/Guardians

When ALWHIV are unaware that they have HIV, families may experience great dread and anxiety. The relationship between parents/guardians and their children may become stressful, and conflicts may arise. Participants indicated their frustrations as follows.

Participant 03 of Shayandima verbalized that, ‘*sometimes as an adolescent, you become angry because you find the truth from outsiders that you are HIV+. Parents are very secretive, especially fathers*.

Participant 02 of Wayeni said, ‘*in 2018, I asked my mother about my chronic pneumonia. She didn’t tell me. At that time, I knew I had HIV. I read it on my file. Then she said pneumonia is a very dangerous condition I must keep on taking my pills. I told her she is lying to me I said I know I had AIDS mom why are you lying to me. My other siblings were not drinking pills like me, which was depressing*’.

However, other participants indicated that parents disclosed their status to their HIV-positive children. Participants expressed the following quotes.

Participant 12 of Vhurivhuri said, “*My parents took me to a social worker who disclosed that I am HIV positive. The social worker helped counsel me, but I did not understand initially until I accepted*.

Participant 06 of Madombidzha said ‘*disclosure is not easy, but my parents told me that I should continue taking my medications and do as nurses and doctors said. My father told me,’ Don’t worry, my boy, you will grow up and understand one day*’.

Participant 02 from Wayeni shared, ‘*Mom took me to my granny’s house. She said, my son, please forgive me your father, and I are HIV+. I was afraid to talk all these years. My boy, please forgive me I was just protecting you*’.

Participant 14 from Muila said, ‘*They should have thought long ago that I will not always remain a child; one day I will grow, and the truth will come out*.

In this study, the researcher noted that adolescents who found out their actual status alone were angry and frustrated and wanted their parents to be accountable. Some ALWHIV defaulted their medications for weeks in need of clarity from parents after they became aware of an accurate diagnosis or their status. Some value themselves as nothing and do not see the value in life anymore. However, later, they found peace to forgive their parents and continue with life.

## 4. Discussion of Findings

ALWHIV were part of the ART programme, had taken ARV for over six months, and disclosed their HIV status. The findings revealed they experienced a poor understanding of the disease condition, improved physical health when adhering to ARV treatments, challenges related to sexual maturity and intimate relationships, and lack of disclosure of HIV+ status by parents. Adolescents who go through a smooth transition can manage their HIV care independently. The study’s findings revealed that ALWHIV with vertically acquired HIV were familiar with the illness and were referred to as being sick, not knowing the exact cause of that. Before knowing their HIV status, participants cited being constantly ill. They thought being HIV positive meant they were bewitched; they had to visit traditional health practitioners for treatment. However, when taking traditional medicine, no physiological improvements were noted. When adolescents do not comprehend the nature and treatment of a chronic illness, they cannot manage it independently [34]. According to Ondwela et al. [35], the idea that conventional treatment can cure HIV is entirely unsupported by science. Witchcraft has frequently been cited as the primary source of HIV and AIDS [36].

Post enrolment on the ART programme, participants indicated strategies to adhere to their ARVs where family members supported and reminded them to adhere. Reminders about treatment frequently benefit adolescents (such as pill boxes, beepers, or timers) [37]. The World Health Organization [38] states that adolescents’ capacity to exhibit fundamental health-care awareness, awareness of their status, understanding of their disease, and developing their own health management abilities are indicators of their readiness for transition. Other participants experienced serious challenges because their bodies were not responding to the medications taken (treatment failure). The treatment regimen was changed as the viral load remained high. Bure et al. [7] stated that genetic variations in the viruses that are chosen by ART medications can lead to drug failure. Adolescents and young adults have considerably greater rates of ARV treatment failure and the emergence of medication resistance, Hassan et al. [39], which could indicate that they are not taking their medications as prescribed, whereas it is documented that the use of ART helps to restore the immune system and maintain viral load at undetectable levels, thus contributing to an increase in quality of life [40]. Concerns about clinical and treatment adherence have been highlighted globally as ALWHIV grow older and take greater responsibility for their medication [3,41]. Switching to second-line ART regimens in a timely manner should be advocated.

Like all adolescents, ALWHIV need love and can be attracted to the opposite sex. They must adhere to consistent condom use and disclose their status to their partner. They understand their responsibility to protect their partners but fear that disclosure would lead to rejection or stigmatisation. HIV disclosure may be essential for HIV prevention and access to medical care and treatment, according to research by Dennison et al. [32]. In addition, hiding an HIV+ status puts one’s risk reduction at danger and puts their spouse or other sexual partners at risk of contracting the virus if they are not already infected [42].

Our interviews with ALWHIV showed that parents/guardians never disclosed that they had vertically acquired HIV. Some ALWHIV were angry and frustrated at growing with the disease. It was mentioned that parents were afraid to disclose to the child, were secretive about HIV issues, or the child may not comprehend the context. The delayed disclosure of a child’s HIV-positive status during the transition to adolescence affects their treatment adherence, causing a high amount of unsuppressed viral load [43]. Some adolescents were told that HIV is a family issue that should not be explained to others. Due to this, ALWH decided not to tell their sexual orientation to friends or lovers [2]. Some participants disclosed that their caregivers had warned them against telling anyone outside the family about their HIV status. Their HIV status was to be kept a secret within the family. Secrets appeared to be an everyday part of participants’ lives as it was difficult for them to trust what others may do with the information about their HIV status. According to Luseno et al. [44], children do not handle being lied to well, especially by those in positions of trust, such as their parents or guardian, doctors, and health professionals. When ALWHIV discovered they had been misled about their HIV status, they said they felt enraged and resentful. The importance of disclosure showed that families with HIV-unaware children can experience great dread and distress [32]. The relationship between parents/guardians and their children may become stressful, and conflicts may arise. A study by Jonkelowitz et al. [45] attested that disclosure opens the possibility of joining support groups with others who fully disclosed. ALWHIV will gain from the knowledge, assistance, and sense of community by attending support groups. This also reduces the chance of adolescents asking difficult questions of their parents or guardians.

## 5. Limitations of the Study

The study was conducted in Limpopo Province within the two selected districts, and with only 27 ALWHIV, the results cannot be generalized to other districts.

## 6. Recommendations of the Study

During the transition, ALWHIV need strong support (physical, emotional, psychological, financial, spiritual, etc.) from healthcare providers, parents, and their families as they move towards independence and medication adherence. Adolescent transition requires strengthening the application of age-appropriate, individualized case management at all sites and developing welcoming environments in the family, peer, and medical communities. A thorough and multidisciplinary programme that tackles their unique concerns is necessary. For sexually active adolescents, the therapeutic education programme promotes condom use, advocates for abstinence, and lowers the number of sexual partners. In addition to follow-up, significant psychosocial support is provided, especially for the most vulnerable, such as teenagers and orphans.

## 7. Conclusions

The study findings showed that HIV status was not disclosed to ALWHIV at the early adolescent stage. They also lacked a clear understanding of disease progression and non-adherence to ART and performed inappropriate practices that could increase the risk of HIV infection. Parents kept their status secret, and positive HIV status was not disclosed to them, which risked transmitting the virus and re-infection if engaging in unprotected sex.

## Figures and Tables

**Table 1 children-10-00798-t001:** Profile of ALWHIV (n = 27).

Demographic	Category (Variable)	Number	Percentage (%)
Age	12–15 16–19	4 23	15% 85%
Gender	Male Female	9 18	33% 67%
Ethnicity	Venda Tsonga Sotho	18 7 2	67% 26% 7%
Educational level	Primary Secondary Tertiary Drop-out	0 25 0 2	0 93% 0 7%
Employment status	Employed Unemployed	2 25	7% 93%
Disclosure of HIV status	Male Female	0 1	0 4%
Non-disclosure of HIV status	Male Female	9 17	33% 63%

**Table 2 children-10-00798-t002:** Themes reflecting experiences of ALWHIV during the transition to adulthood.

Theme Number	Theme Description
Theme 1	Poor understanding of the disease condition
Theme 2	Improved physical health experienced as associated with adherence to treatments
Theme 3	Challenges related to sexual maturity and intimate relationships
Theme 4	Lack of disclosure of HIV+ status by parents

## Data Availability

The corresponding author can provide some of the raw data used to support the findings that are contained in the article upon request. This manuscript was derived from a doctoral thesis.

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
