# Peer review of "Growing Up with HIV: Experiences of Transition from Adolescence to Adulthood at Selected Primary Health Facilities in Limpopo Province, South Africa"

_children, 2023, doi:10.3390/children10050798_

Round 1
Reviewer 1 Report
The authors have made an interesting attempt at “Growing up with HIV: Experiences of transition from adolescent to adulthood at selected primary health facilities, Limpopo Province.” The manuscript is interesting; however, the authors need to justify the scientific writing manuscript. Some of the general comments are provided below:
1. Authors should improve the introduction to elaborate their work with related examples from the literature.
2. Did the authors conduct a pilot study to validate the clarity and comprehensibility of survey questions, if yes, what is the test score of Cronbach's alpha?
3. What is the representative minimum sample size for the population required for statistical analysis?
4. The authors did not explain the exclusion criteria of participants for this review.
5. Some paragraphs in the introduction need references (for example. lines 43-52)
6. In table 1: What does the author mean by disclosure by gender” and “not disclosed”?
7. Table 1: “ not disclosed”, how do they have 53 participants here? (Male 27, female 26) when there are only 27 participants.
8. The authors should have made another table to explain the four themes, as they have shared a few examples from their participants, and they did not explain the results of all participants in each theme.
9. In addition to the above point, the health conditions of the participants should also be explained in aa organized form.
Author Response
All suggested comments have been addressed in the manuscript, highlighted in red, and referred to the line number.

Reviewer 2 Report
I stopped my review before finishing reading the contents of Introduction because of too many grammatical errors. The authors should review their manuscript and sent the manuscript for English editing.
Title: The name of the country should be added.
Abstract
1. “ALWHIV”: full spelling should be presented.
2. Selected primary health-care facilities
3. Tesch’s
4. The words in Abstract were Inconsistent font size.
5. This sentence should be corrected. “A comprehensive and multi-sectorial programme that is adolescent friendly to be implemented to address various ALWHIV challenge.”
Introduction
“PMTCT”: full spelling should be presented.
“In the past five years, from 2010 to 2015, it has been estimated that 1.3 million of these had been avoided [2].” it should be corrected.
Typo: g [1/2,4].
Line 48: Dahourou DL, Gautier-Lafaye.
Author Response
All the reviewer's suggestions are corrected; the authors reflected that in the manuscript, highlighted in red, and indicated the line number.

Reviewer 3 Report
The manuscript consists of total 9 pages, including 1 table and the list of total 37 literature references. The original material-based article presents the results of the Authors' study on from the birth-on HIV-positive youths experiences associated with changing their social roles from children to adults, stressing the bases of its influence on their anti-retroviral therapy adherence. The article comes from one of countries of Africa where the HIV problem is especially important for local communities and the healthcare systems suffer from many deficiencies, often resulting from overall financial limitations - so publishing the article has increased value for topic-interested Readers as such literature is still rather scarce. The article follows in general a recognizable logical structure and the line of argumentation is easy to follow. The English language quality used by Authors is generally acceptable although several corrections are still required.
The Abstract section mirrors both structure and key contents of the main text of the manuscript. The ALWHIV, ARV abbreviations need to be explained while used for the first time in this section.
The Introduction provides the context and reason for the study. The Authors may consider screening the text for multiple times-repeated information and remove the repetitions where necessary - for smoother reading experience. All abbreviations need to be explained while used for the first time in the main text of the article, e.c PMTCT line 36, ALWHIV line 40, for the sake of ease of reading.
The sentence line 72-75 is too long and its meaning is unclear "Since 72 ALWHIV are still developing physically, psychologically, and sexually, they encounter particular developmental, psychological, and sex issues Lowenthal et al [13, frequently in the context of stringent parental consent rules and regulations [14]."
typos noted, e.c. : line 43 [1/2]
The Materials and methods section is clear and informative enough.
The Results named Presentation of findings section is structured for better clarity, consistent with the declared study methodology, supported by a table - here the last row "not disclosed" needs clarification as the numbers 27+26 do not add up here to n=27 number of participants. The citations from the actual participants subjective disclosures are especially Reader-appealing and support additionally the Authors' line of argumentation.
The Discussion section places the own findings of the Authors into the broader context of previously published knowledge.
The line 266-267 sentence needs clarification "The scientific basis for curing HIV with conventional medicine, according to Ondwela et al [28], appears to be a complete myth."
The section Recommendations shall contain the Authors' views and opinions, thus it shall not contain literature references.
The literature references are numerous and recent enough.
The Authors may consider including into Their article also the following mentions adding to the context of their study:
- the problem of HIV mutations, that are more common in case the adherence to the treatment regimen is poor as in e.c. https://doi.org/10.3390/v7020590
- the HIV infection as psychological challenge as in e.c. https://doi.org/10.3390/children10020405 https://doi.org/10.3390/ijerph20042996 https://doi.org/10.3390/children9121989 https://doi.org/10.3390/children9121955 https://doi.org/10.3390/ijerph192214710
- the exposure to HIV as psychological challenge as in e.c. https://doi.org/10.3390/ijerph20032499
- unprotected sexual practices as obstacle in HIV control as in e.c. https://doi.org/10.3390/adolescents3010004 https://doi.org/10.3390/ijerph192113763
Author Response
All the reviewer's suggestions have been considered, this is highlighted in Red and line number indicated.

Round 2
Reviewer 1 Report
The authors have modified the manuscript and now it is acceptable for the publication.
Reviewer 2 Report
The authors have revised their manuscript based on the reviewer's suggestions. I would like to suggest the editors accepting it for publication.